# Growth Inhibition and Additive Effect to Antimalarial Drugs of *Brucea javanica* Extracts on Asexual Blood-Stage *Plasmodium falciparum*

**DOI:** 10.3390/pathogens14070646

**Published:** 2025-06-30

**Authors:** Niwat Kangwanrangsan, Gamolthip Niramolyanun, Chonnipa Praikongkatham, Pathanin Chantree, Pongsakorn Martviset, Viriya Pankao

**Affiliations:** 1Department of Pathobiology, Faculty of Science, Mahidol University, Bangkok 10400, Thailand; niwat.kan@mahidol.ac.th (N.K.); gamolthip.nir@gmail.com (G.N.); chonnipa.pr@gmail.com (C.P.); 2Department of Preclinical Sciences, Faculty of Medicine, Thammasat University, Pathumthani 12120, Thailand; pathanin@tu.ac.th (P.C.); pong_m@tu.ac.th (P.M.)

**Keywords:** malaria, medicinal plant, additive effect, artesunate, chloroquine

## Abstract

Malaria is a parasitic infectious disease that is endemic in many tropical countries. Even though several effective antimalarial agents have been implemented, treatment failure still occurs, and malaria continues to cause neurological complications and death, particularly in severe or drug-resistant cases. Hence, novel therapeutic agents with distinct mechanisms of action, as well as alternative chemical compounds that can overcome resistance, are still needed to improve malaria therapy. This study aimed to investigate the antimalarial activities of *Brucea javanica*, a tropical plant extracts against *Plasmodium falciparum*, the major species associated with severe malaria. In this study, malaria parasites were treated with plant extracts using single and co-incubation methods, along with artesunate and chloroquine, and their inhibitory effect on parasite development was determined by microscopy. The results show that all tested doses of the extracts that effectively inhibited malaria parasites did not cause hemolysis of red blood cells (RBCs). The root extract (RE) and fruit extract (FE) inhibited parasite growth at IC_50_ values of 0.41 ± 1.14 µg/mL and 0.26 ± 1.15 µg/mL, respectively. These plant extracts significantly interrupted malaria development at the ring stage, as presented by a reduction in the conversion rate to trophozoites and schizonts. The defective parasites treated with plant extracts were characterized by nuclear clumping, leading to pyknotic cell death. Moreover, RE and FW extracts elicited an additive effect with artesunate and chloroquine, significantly reducing IC_90_ levels for the inhibition of parasite development. In conclusion, *B. javanica* extracts inhibited the asexual blood-stage development of malaria parasites. They distinctively show the additive effects of ATS and CRQ, elucidating their potential for further studies on novel formulas of antimalarial drug regimens.

## 1. Introduction

Malaria is one of the most recognized infectious diseases globally, caused by the Apicomplexa protozoan *Plasmodium*, transmitted to humans through the bites of infected female *Anopheles* mosquitoes. Among the five species that infect humans—*P. falciparum*, *P. vivax*, *P. ovale*, *P. malariae*, and *P. knowlesi*—*P. falciparum* is responsible for the most severe cases and the majority of malaria-related deaths globally [1]. According to the report by the World Health Organization, malaria continued to impose a significant public health burden, with an estimated 263 million cases and 597,000 deaths in 2023. The burden was particularly high in sub-Saharan Africa, Southeast Asia, and parts of South America, where transmission was driven by the regional presence of competent mosquito vectors [2].

The complex life cycle of *P. falciparum* includes both asexual and sexual stages. After transmission via a mosquito bite, sporozoites travel to the liver, where they undergo asexual replication, known as exoerythrocytic schizogony, to produce thousands of merozoites. These merozoites are released into the bloodstream, invade red blood cells (RBCs), and initiate the erythrocytic cycle. Inside RBCs, the parasite develops from a ring stage to a trophozoite and matures into a mature schizont, eventually rupturing the host cell and releasing new merozoites [3]. The repeated rounds of asexual intraerythrocytic replication lead to clinical symptoms such as high fever with severe shaking chills and anemia, contributing to disease severity [4,5]. A subset of parasites differentiates into gametocytes and is further taken up by mosquitoes during a blood meal to continue the sexual stage development, thereby completing the transmission cycle. The obtaining of blood gametocytes and the successful development of the mosquito stage could give rise to the infective stage to further new malaria patients [3,6].

A hallmark of *P. falciparum* pathogenesis is the sequestration of infected RBCs in the microvasculature of organs. This is mediated by parasite-derived proteins, such as *P. falciparum* erythrocyte membrane protein 1 (*Pf*EMP1), which could bind to host adhesion molecules, namely ICAM-1, CD36, and the endothelial protein C receptor [7,8]. This cytoadherence could obstruct the small vessels, causing several severe conditions, including cerebral malaria, renal failure, metabolic acidosis, and occasionally death [9].

The emergence and spread of drug-resistant *P. falciparum* strains have become a significant obstacle to malaria control and elimination [10,11]. Chloroquine (CRQ) was once the primary malaria drug, but *P. falciparum* developed widespread resistance. However, it remains effective against *P. vivax* and *P. malariae* in certain regions [12,13]. Artemisinin-based combination therapy (ACT), which pairs fast-acting artemisinin derivatives (e.g., artesunate, dihydroartemisinin) with longer-acting partner drugs (e.g., piperaquine, lumefantrine, mefloquine), was subsequently developed as the first-line treatment [14,15]. However, partial resistance to artemisinin and the decreasing efficacy of partner drugs was recorded, especially in the Greater Mekong Subregion, including Thailand, Cambodia, and Myanmar [16,17,18,19]. Moreover, artemisinin resistance in Africa was recently identified, with an increasing prevalence of the PfKelch13 mutation. These artemisinin-resistant PfKelch mutants were identified in low transmission in several countries, including Rwanda, Uganda, Ethiopia, Sudan, Kenya, and the eastern Democratic Republic of the Congo (DRC), indicating the spread of artemisinin-resistant parasites over large distances [20,21]. This evidence raised serious concerns about future treatment failures.

Given these challenges, there is a pressing need for novel antimalarial agents that can overcome the therapeutic inefficiency and control the disease. One of the promising sources of new therapeutics is traditional medicinal plants. *Brucea javanica* (L.) Merr., a small tree in the Simaroubaceae family, has a long history of use in traditional medicine across Asia and Oceania [22,23]. The crude extracts and active ingredients of this plant have been studied in vitro and in vivo for their therapeutic effects in several diseases, dating back to the Qing Dynasty of China (1368–1644 AD) [24,25,26]. *B. javanica* crude extracts have been known for their anticancer properties, which have been proposed to establish several mechanisms in various cancers, including cervical cancer [25], breast cancer [27,28], hepatocellular carcinoma [29], and pancreatic cancer [30]. Moreover, antiviral [31], anti-inflammatory [32], and neuroprotective properties [33,34] have also been reported. Interestingly, *B. javanica* exhibited antimalarial activity in both in vitro and in vivo models, including drug-resistant *P. falciparum* and *P. berghei* [35,36,37,38]. Several extraction procedures have been established to isolate various active constituents, such as bruceine A and bruceine B, illustrating distinguished results [39]. However, most of these studies had been reported only for screening results, with minimal mechanistic insights into how these extracts affect intra-erythrocytic parasite development [40]. Moreover, while a few reports suggested potential support for chloroquine treatment [37], there have been no studies to date that have explored its combinations with artemisinin-based therapies or compared the efficacy of extracts from different plant parts.

In this study, we evaluated the antimalarial activity of *B. javanica* extracts—specifically ethyl acetate and aqueous fractions derived from leaves, fruits, and roots—against *P. falciparum*. We also assessed how these extracts, alone and in combination with artesunate (ATS) and chloroquine (CRQ), affected parasite morphology and development during the blood stage. The results of this work could provide novel insights into the therapeutic potential of *B. javanica* and explore its role as a complementary strategy for malaria treatment.

## 2. Materials and Methods

### 2.1. Plant Collection and Identification

*Brucea javanica* was obtained from its natural source in the Ta-Kua-Pa district of Phang Nga Province, Thailand. The plant species was identified and deposited in the Herbarium (number SKP 178 02 10 01) by the Faculty of Pharmaceutical Sciences, Prince of Songkla University, Hat Yai, Songkhla, Thailand.

### 2.2. Preparation of B. javanica Extract

The leaves, fruits, and roots of *B. javanica* were dried and ground into powder. Each part was extracted using two different solvents, including ethyl acetate and water, in a ratio of 1:10 (weight by volume). They were mixed and orbital shaken at 180 rpm/min for 6 h at room temperature. The mixtures were separated by centrifugation at 1000× *g* for 5 min, and the supernatants were filtered through Whatman filter paper No. 1 before being collected. The pellets were dissolved in the solvent and shaken again at 180 rpm/min for 16–18 h at room temperature. Each part of the crude extracts from the first and second times was pooled together. The crude extracts from leaves, fruits, and roots, obtained using ethyl acetate, were labeled as LE, FE, and RE, respectively. They were evaporated using a Rotavapor R-200 rotary evaporator (Buchi, Flawil, Switzerland) at 40 °C and then freeze-dried by Super Modulyo Freeze Dryer (Thermo Fisher Scientific, Waltham, MA, USA) for 24 h to obtain a solid residue. The crude extracts from leaves, fruits, and roots, obtained by water, were designated as LW, FW, and RW, respectively. They were finally freeze-dried, as mentioned, to obtain a solid residue. The percentage of crude extract yield was calculated for the next experiment.

### 2.3. In Vitro Culture of P. falciparum

Blood stages of *P. falciparum*, NF54 strain, were cultured with O-type human red blood cells (purchased from Thai Red Cross, Bangkok, Thailand) at 5% hematocrit in the complete culture medium consisting of RPMI-1640 medium with L-Glutamine (Gibco, Grand Island, NY, USA), 2 g/L sodium bicarbonate (Mallinckrodt, Dublin, Ireland), 25 mM HEPES (Sigma-Aldrich, Burlington, MA, USA), 2 g/L dextrose (M&B, Birmingham, Great Britain), 50 mg/L hypoxanthine (Sigma-Aldrich, Burlington, MA, USA), 10% heat inactivated human serum type A positive, and 10 mg/L gentamicin under the condition of 37 °C with mixed gas (90% N_2_, 5% O_2_, and 5% CO_2_) [41]. The parasite was synchronized by 5% sorbitol (PanReac AppliChem ITW Reagents, Barcelona, Spain) to enrich for ring-stage parasites [42]. Two consecutive sorbitol treatments were performed to achieve tight synchronization. The synchronous parasites were confirmed to be at the 6–12 h ring stage under the microscope. The parasite number was then adjusted to 1% parasitemia for further growth inhibition and co-incubation assays.

### 2.4. Antimalarial Activity Test of B. javanica Extracts by Growth Inhibition Assay

*B. javanica* extracts were tested for toxicity to human red blood cells (hRBCs) by in vitro hemolysis assay [43]. Briefly, the O-type hRBCs at 5% hematocrit were incubated with plant extracts at the final concentrations of 0.01, 0.1, 1, 10, 100, and 1000 μg/mL under the temperature of 37 °C for 0, 1, 6, 18, and 24 h, compared to the positive and negative control, 0.1% Triton X in PBS, and 0.5% DMSO in PBS, respectively. The extracts that showed no toxicity to hRBCs were then tested for their antimalarial activity by growth inhibition assay [44]. Briefly, the synchronized ring stage of *P. falciparum* was cultured at 1% parasitemia in 2.5% hematocrit and then incubated with the plant extracts at the concentrations of 0.01, 0.1, 1, 10, and 100 μg/mL on a 96-well plate (SPL Lifescience, Pocheon-si, Republic of Korea) for 26–28 h in a hypoxic chamber containing mixed gas (90%N_2_, 5%O_2_ and 5%CO_2_). In parallel, 0.05% DMSO was concurrently used with artesunate (ATS; Atlantic Pharmaceutical, Bangkok, Thailand) and chloroquine (CRQ; Sigma-Aldrich, Burlington, MA, USA) at serial concentrations as negative and positive controls, respectively. Each treatment condition was performed in duplicate wells of three independent experiments for statistical analysis. After the incubation period, samples from each treatment condition were collected to prepare Giemsa-stained thin blood smears. Parasitemia and parasite stages were examined under a microscope to assess the effects of treatment. The percentage of parasite growth inhibition was calculated by comparing the parasitemia of the treated groups with that of the control. The plant extracts that showed high antimalarial activity were then selected for co-incubation with antimalarial drugs in a further experiment.

### 2.5. Co-Incubation of B. javanica Extracts with Antimalarial Drugs

The selected plant extracts were evaluated for antimalarial activity in combination with the known antimalarial drugs artesunate (ATS) and chloroquine (CRQ). Two selected *B. javanica* extracts (one from ethyl acetate extract and the latter from water extract) at the fixed IC_50_ doses were co-incubated with 10-fold serial concentration (from 0.01 to 100 ng/mL) of ATS, and those (0.1 to 1000 ng/mL) of CRQ. The treatments of malaria parasites with ATS or CRQ alone and 0.05% DMSO served as the control groups. The synchronized ring stage parasites were cultured at 1% parasitemia in 2.5% hematocrit and incubated with a combination of selected *B. javanica* extracts and ATS or CRQ on a 96-well plate for 26–28 h at 37 °C in a hypoxic chamber containing a mixed gas (90% N_2_, 5% O_2_, and 5% CO_2_). After incubation, the samples from each well were prepared for thin film Giemsa’s smears to count the number of intact parasites under the microscope. The percentage growth inhibition was calculated by comparing the parasitemia of the treated groups with that of the DMSO control. The experiment was performed in duplicate wells for three independent experiments, allowing for statistical analysis.

### 2.6. Microscopic Analysis of P. falciparum from Growth Inhibition and Co-Incubation Assay

The samples from each well of the growth inhibition and co-incubation assays were collected to make thin blood films and stained with Giemsa’s solution. To estimate the number of intact parasites, Giemsa smears were observed and counted under an Olympus uplight microscope model BX53 (Olympus, Tokyo, Japan) for parasites residing in RBCs with normal development to trophozoite and schizont stages, characterized by normal nuclei and cytoplasmic patterns at high magnification. In each treatment condition, five independent slides were counted for 10,000 erythrocytes (approximately 30 fields at 100× objective lens magnification), to calculate parasitemia compared to the control. The results were then converted to % growth inhibition using the following equation.% growth inhibition = 100 − ((% parasitemia of sample)/(% parasitemia of control) × 100)

The antimalarial activities of all the extracts were represented by the IC_50_ (half-maximum inhibitory concentration) values. These values were calculated using Microsoft Windows software. The graphs were plotted with the percentage of inhibition on the *Y*-axis and the extract concentration on the *X*-axis, using a linear regression equation. The additive efficacies of *B. javanica* extracts to antimalarial drugs were represented by the IC_90_ values of ATS and CRQ co-incubated with plant extracts compared to ATS or CRQ treatment alone. Parallel with counting the parasite numbers, the parasite stages, including ring, early trophozoite, trophozoite, and schizont, were also identified to evaluate the distribution of developmental stages. A single observer performed the parasite count. The second observer cooperated in confirming parasite staging and morphological defects. The conversion rates from rings to trophozoites and from rings to schizonts were calculated from the number of trophozoites and schizonts specifically counted after 26–28 h of plant extract treatment compared to the number of rings at 0 h. Finally, the morphological defects of the parasites were determined and photographed using CellSens Imaging Software version standard 4.3.

### 2.7. Statistical Analysis

The significance of variability between the data from each group was expressed as mean ± standard deviation (SD). The statistical analysis applied a Student’s *t*-test using the SPSS 13.0 program for Windows. Statistical differences were considered significant when *p* ≤ 0.05.

## 3. Results

### 3.1. B. javanica Extracts Inhibit P. falciparum Survival

From the hemolysis assay, the leaf and root extracts prepared with ethyl acetate (LE and RE) at a concentration of 1000 μg/mL were found to be toxic to normal red blood cells. However, the other extracts (FE, LW, FW, and RW) did not exhibit hemolysis at any dose or incubation time point (Appendix A). To avoid hemolysis in the growth inhibition assay, the optimal concentrations of all plant extracts used were selected from 0.01 to 100 μg/mL.

To allow parasites to develop beyond the ring stage, synchronized *P. falciparum* cultures (6–12 h rings) were incubated with or without *B. javanica* crude extracts for 26–28 h. Parasitemia in the treatment groups was then compared to that in the control groups to calculate the percentage of growth inhibition. Artesunate (ATS) and chloroquine (CRQ), including the plant extracts, could damage and reduce the numbers of intact parasites (normal morphology and developmental stage) in a dose-dependent manner. With this, ATS and CRQ exhibited the IC_50_ of 4.94 ± 1.08 and 55.02 ± 1.06 ng/mL, respectively (Figure 1A,B). Focusing on *B. javanica* crude extracts, a more growth-inhibitory effect was found with increasing concentrations of both ethyl acetate and water extracts. According to ethyl acetate extracts, RE presented the growth inhibition level at IC_50_ of 0.41 ± 1.14 μg/mL, followed by LE at 1.82 ± 1.12 μg/mL, and FE at 27.32 ± 1.16 μg/mL (Figure 1C). For water extracts, FW showed the inhibitory effect at IC_50_ of 0.26 ± 1.15 μg/mL, RW at 1.83 ± 1.13 μg/mL, and LW at 11.24 ± 1.08 μg/mL (Figure 1D). The data distinctly demonstrated that all the crude extracts from this plant obtained the potential for antimalarial activity. Statistically, RE and FW were the two most effective extracts, showing remarkable potential for further investigation. In parallel with the microscopic detection, the percentage growth inhibition was also assessed using real-time RT-PCR targeting 18S rRNA expression (Appendix A). Although the trends observed by both methods were consistent, real-time RT-PCR showed relatively higher levels of parasite signal under the same treatment conditions, likely due to the higher sensitivity of this molecular technique. Moreover, they both presented different advantages and disadvantages. A thin blood film was preferred for observing and classifying the parasite stage distribution, including morphological changes, which could predict the mechanism of the associated agents. However, this method could not completely express whether the agents can kill the parasite. On the contrary, the real-time RT-PCR method could only quantify live parasites. It has more sensitivity than a thin blood smear, but the structural changes could not be observed.

### 3.2. B. javanica Extracts Arrested the Development of P. falciparum

To understand the developmental stage affected by plant extracts, the microscopic analysis was further focused on the distribution of the parasite stage. After 26–28 h of incubation, almost all control (non-treated and 0.05% DMSO-treated) ring-stage parasites developed normally into trophozoites and schizonts. A large proportion of defect/death parasites, with some remaining in the ring stage, was observed at 100 ng/mL of ATS. At 10 ng/mL of ATS, treated parasite survival was composed of a significant proportion of early trophozoite and trophozoite stages. Parasites treated with 100 and 1000 ng/mL of CRQ showed limited growth inhibition, but exhibited delayed development, remaining mostly at the ring and early trophozoite stages with only a small proportion of defective or dead parasites (Figure 2A). The distributions of parasite stages in the group of low concentrations (0.01 and 0.1 μg/mL) of RE and FW treatment showed a similar pattern with those of normal development observed in the control groups. About 80–90% of parasites treated with 1, 10, and 100 μg/mL of these extracts delayed their growth at the ring stage with small proportions of early trophozoites and defective parasites (Figure 2B). In consideration of parasite retardation, the conversion rates of parasites from rings trophozoites and rings to schizonts were analyzed. Remarkably, the conversion rates from the ring stage to trophozoite and from the ring stage to schizont absolutely ceased in medium and high doses of the plant extracts (Figure 2C). These microscopic observations distinctly reveal that RE and FW extracts arrested parasite growth at the early stage of development. Since it was clearly recognized that the development of parasites treated with RE and FW extracts was halted, we mimicked a pulse-chase experiment to study the dynamics of the developmental process. After 26–28 h of incubation with or without plant extracts, all the parasites in the different treatment conditions were washed with incomplete medium before continuing with standard malaria parasite culture without any additional treatments for 24 h. The results show that the groups with no treatment and lower concentrations (0.01 and 0.1 μg/mL) of RE and FW extracts could continue their growth, as indicated by an increase in parasite numbers from 1% to 3% parasitemia. Interestingly, the development of parasites initially treated with 1, 10, and 100 μg/mL of plant extracts was not resumed but remained in the ring stage. The permanent blocking of the developmental process of malaria parasites implied the irreversible inhibition of *B. javanica* extracts (Appendix A).

### 3.3. Morphological Changes in Malaria Parasite After Treatments with Antimalarial Drugs and B. javanica Extracts

Start from the ring stage at 0 h (Figure 3A) of growth inhibition assay; intra-erythrocytic parasites developed into typical trophozoites and schizonts, which were indicated by nucleus and cytoplasmic staining, as found in DMSO treated group (Figure 3B). A significant number of defected parasites was observed in 10 and 100 ng/mL of the ATS treated group with remarkable hemozoin clumping (aggregation of the hemozoin pigment appearing from golden brown to dark brown or black color in the parasite’s cytoplasm) and pyknotic cell death (nuclear clumping), respectively (Figure 3C). Additionally, accumulations of the parasite’s digestive vacuoles were also observed on top of hemozoin clumping in 100 and 1000 ng/mL of the CRQ treated group (Figure 3D). On the other hand, the morphological change that was frequently found on parasites treated with RE and FW extracts was nuclear clumping on the ring stage (Figure 3E,F). It has been noted that there was no hemozoin clumping found in any groups of parasites treated by plant extracts. It could be inferred that the plant extract obstructed the parasite growth in different partways of ART or CRQ, and could generate the additive effect to ATS and CRQ.

### 3.4. Additive Effect of B. javanica Extracts on Antimalarial Drugs

Designed for additive effect treatments of plant extract on ART and CRQ, the *B. javanica* RE and FW were achieved at constant concentrations of the IC_50_ values, in combination with graded serial dilutions of ATS and CRQ. The experiment was divided into four groups, including ATS+RE, ATS+FW, CRQ+RE, and CRQ+FW, compared with the positive controls, ATS or CRQ alone. Since the inhibitory effect of the combination treatment dramatically increased to more than 70% from the lowest dose of antimalarial drugs, the inhibitory effect was represented by the IC_90_ value. The additive efficacy of RE and FW extracts on ATS is shown in Figure 4A. ATS+RE and ATS+FW exhibited growth inhibition at IC_90_ concentrations of 12.92 ng/mL and 7.52 ng/mL, respectively, which was more effective than that of ATS alone at an IC_90_ concentration of 32.40 ng/mL. An improvement in the growth inhibition from CRQ was also observed in combination treatment by CRQ+RE and CRQ+FW (IC_90_ of 84.14 ng/mL and 96.18 ng/mL) when compared to the treatment with CRQ alone (IC_90_ of 183.40 ng/mL) (Figure 4B). Notably, when RE and FW extracts were applied at their IC_50_ concentrations in combination with low doses of ATS (0.01, 0.1, and 1 ng/mL) or CRQ (0.1, 1, and 10 ng/mL), the growth inhibition reached approximately 70–80%, a marked increase compared to the 5–10% inhibition observed with ATS or CRQ alone at the same low doses. This significant enhancement supports an additive effect of *B. javanica* extracts in potentiating the efficacy of standard antimalarials, even at subtherapeutic drug concentrations.

The parasite’s morphological change observed in the combination treatment of ATS+RE and ATS+FW (Figure 4C,D) was nuclear clumping (at the level of 100 ng/mL of ATS). There was no detection of hemozoin clumping as previously found in the ATS treatment alone. This is supported by microscopic observation that the treated parasite ceased in the ring and early trophozoite stages. This suggests that the primary effect of plant extracts was on top of the damage caused by ATS. A comparable phenomenon was observed in the combination treatment between CRQ and plant extracts (Figure 4E,F). Only nuclear clumping was detected without hemozoin clumping or vacuole accumulation in the cytoplasm of treated parasites.

## 4. Discussion

The malaria parasite poses a significant health problem in several countries worldwide because transmission is facilitated by mosquito vectors, specifically *Anopheles* [3]. Moreover, the symptoms that develop after infection with particular species, such as *P. falciparum*, can sometimes be virulent [1]. The clinical pathophysiology of *falciparum* malaria mainly involves the blood stages. The hemolysis of infected red blood cells (iRBCs) that released daughter merozoites caused sudden high fever. Moreover, the aggregation of iRBCs in the capillaries can be the cause of severe malaria, such as cerebral malaria, renal failure, and respiratory failure [4]. Hence, eliminating the blood stages could appropriately manage the infected patients.

Apart from the complexity of malaria biology, the most significant concern regarding *P. falciparum* is drug resistance [2,17,19]. Artemisinin-combination therapy (ACT), the most effective therapeutic regimen for *falciparum* malaria, was introduced more than 20 years ago, with several formulations that potentiate its killing efficacy, thereby reducing the incidence of severe cases and mortality. However, drug resistance is still increasingly reported in several regions, particularly Southeast Asian countries, including Thailand [17]. To address this issue, several studies have demonstrated that combining antimalarial drugs, especially artemisinin derivatives, with novel compounds may be a viable solution or at least delay the parasite’s development of drug resistance [15,45].

However, before combining these compounds or extracts, their antimalarial activities should be thoroughly studied. Hence, this present study aims to investigate the antimalarial activities of *B. javanica* extracts, which have been primarily reported for their antimalarial properties [37,38,39,40] and tested for their combination efficacy with *falciparum* malaria when used in conjunction with the standard drugs, ATS and CRQ, also originating from a local plant.

We extracted the bioactive constituents of *B. javanica* leaves, fruits, and roots by using two solvents, ethyl acetate and water, in which the different ratios of active constituents were expected. The previous studies reported that *B. javanica* has major constituents consisting of quassinoids (e.g., bruceine, bruceanic acids, javanicolide, javanic acids, yadanzioside, bruceoside, and quassilactone), alkaloids (e.g., bruceolline and bruceacanthinoside), and triterpenoids (e.g., brujavanone) that could be found in different proportions when extracting the other parts of the plant [26]. The major constituents in the *B. javanica* fruits are quassinoids, including bruceine H, J, M, yadanzioside A, B, C, D, E, F, G, M, S, bruceantin, bruceoside A, quassilactone A, and B. In contrast, the root contains different quassinoids, including brujavanol A and B. The leaves have limited information on their active ingredients but are expected to have some triterpenoids [46].

Every extract demonstrated antimalarial activity, but the two most outstanding (based on calculated IC_50_) were RE (ethyl acetate root extract) and FW (water fruit extract), which inhibited parasite growth at low concentrations. Although the antimalarial effect of RE and FW in this experiment was revealed at 20–100 times lower concentrations than those of CRQ and ART, which showed their inhibitory effect at the nanogram level, the IC_50_ values of RE and FW, which were in the sub-microgram range, could be classified as excellent activity. Among 756 herb extracts reviewed for in vitro antiplasmodial activities, *B. javanica* extracted by methylene chloride was categorized in the group of highest activity together with the crude extracts from *Harungana madagascariensis* and *Quassia africana*, and pure compounds established from *Picrolemma spruce*, *Aspidosperma vargasi*, *Aspidosperma desmanthum*, and *Artemisia annua* [47].

Compared to a previous study, our study reports a comparable efficiency of the fruit extract, with IC_50_ values nearly identical to those of *P. falciparum* extracted with different solvents, including ethyl acetate and ethanol-methanol [40]. This finding suggests that the fruit of *B. javanica* could be developed as an antimalarial drug in the future.

Moreover, our study initially tested the *B. javanica* root extract against the malaria parasite, yielding promising results. The RE extract exhibited a low IC_50_, comparable to that of the fruit extract (FW). Therefore, these two extracts have been selected for use in the other experiments. However, the IC_50_ of ATS and CRQ were still lower than those of RE and FW extracts; but this does not matter, as the antimalarial activities of RE and FW remained within safe doses.

Regarding safety, a hemolysis assay revealed that high concentrations of RE and LE extracts (1000 µg/mL) induced significant hemolysis of erythrocytes. However, the concentration required to cause hemolysis was much higher than the IC_50_ concentrations against *P. falciparum*. A previous report on the water extract of *B. javanica* fruit (resembling FW in our study) treated on peripheral blood mononuclear cells demonstrated a safety dose of less than 30 µg/mL. The extract also elicited cytotoxicity in HepG2, a human hepatocellular carcinoma cell line, with an IC_50_ of 1.56 ± 1.02 µg/mL, which was six times higher than our IC_50_ on *P. falciparum* treatment [48]. These suggest a favorable therapeutic window for the use of extracts, although caution is warranted, particularly for formulations intended for clinical use. It is speculated that the ethyl acetate extraction of roots and leaves may concentrate hydrophobic or amphipathic compounds, including certain quassinoids and triterpenoids, that disrupt erythrocyte membranes [49].

Interestingly, the RE and FW extracts interrupted the development of the asexual blood stages of *falciparum* malaria. Although the extracts could not kill the parasite at a high percentage, the RE and FW effectively blocked parasite development at the ring stage in extremely high proportions compared to the standard drugs. These results suggest that *B. javanica* RE and FW extracts could serve as additive agents to antimalarial drugs such as artesunate or chloroquine, as they may block parasite development and lead to the arrested development of the parasite at the ring stage—a stage known to be highly sensitive to these drugs and thus more easily eliminated [19].

Moreover, the RE and FW extracts demonstrate the morphological alterations in the treated *falciparum* malaria parasites. The condensed cytoplasm as well as the occurrence of a considerable accumulation of hemozoin are significant alterations in the parasite after treatment. The results suggest that not only did it slow down the development of the parasite, but the treatments of RE and FW extracts also illustrated the dysmorphological characteristics of the parasite. Moreover, these results correspond with those of other effective antimalarial drugs, which have been shown to alter the course of the parasite after treatment [50,51,52].

As mentioned in the results, the *B. javanica* RE and FW extracts were unable to significantly kill the parasite by causing the recognized defects. However, they could interrupt the blood-stage development, so the additive effect was expected to be tested. The variations in the concentrations of standard drugs, both ATS and CRQ, were tested with constant concentrations of RE and FW extracts. If the combination is successful, treatment with low concentrations of ATS or CRQ would be effective even at low doses, promoting safer use. Our results demonstrate that the combination of ATS and RE (ATS+RE) could significantly lower the IC_90_ compared to the ATS single treatment by approximately 2.5 times. Interestingly, the combination of ATS and FW (ATS+FW) demonstrates an additive effect, decreasing the IC_90_ by 4.5 times compared to the single treatment of ATS. Additionally, for the CRQ combinations, both CRQ+RE and CRQ+FW combinations reduced the IC_90_ by a factor of two compared with the CRQ control. Further, from the morphological investigations, the combinations of RE and FW with ATS and CRQ demonstrated significant alterations in parasites. These findings suggested that *B. javanica* extracts could act as adjuncts to ATS and CRQ by exhibiting additive effects that improve antimalarial activity. This is the first report demonstrating the additive effects of *B. javanica* extracts in *P. falciparum* treated with ATS or CRQ. Moreover, further studies on the active ingredients of this plant could elucidate the mechanism behind its greater antimalarial effect, supporting the development of novel combination formulas that could be beneficial for treating infected patients and controlling the disease.

## 5. Conclusions

*B. javanica* extracts inhibited the growth of asexual blood-stage *P. falciparum* with low toxicity from the early stages of development. The plant extracts demonstrated irreversible developmental arrest at the ring stage, indicating a potent and lasting effect on the parasites. The combination of plant extracts with antimalarial drugs exhibited an additive effect on both ATS and CRQ. Further study on this herbal extract would benefit supplement or drug development, improving malaria treatment and control.

## Figures and Tables

**Figure 1 pathogens-14-00646-f001:**
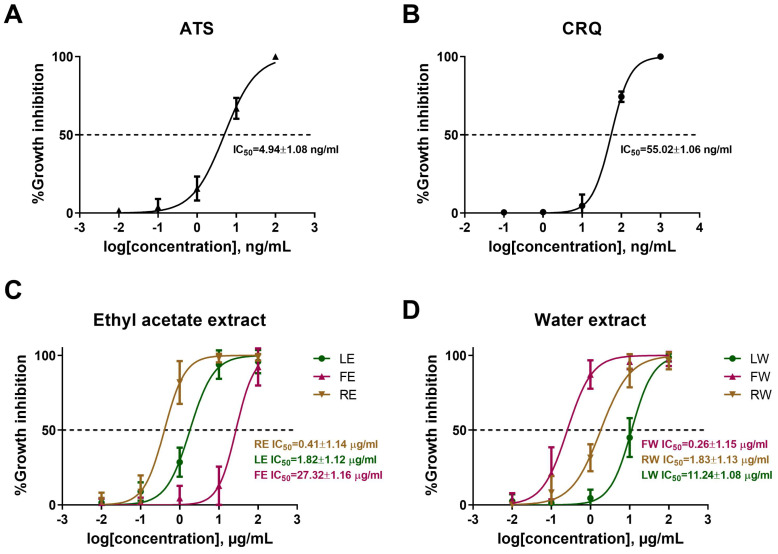
Growth inhibition of *P. falciparum* by *B. javanica* extracts compared to artesunate (ATS) and chloroquine (CRQ). Dose-response curves represent the relationship between the concentration (in logarithmic scale) and the percentage of growth inhibition of the ATS in (**A**), CRQ in (**B**), *B. javanica* (leave, fruit, root) extracted by ethyl acetate (LE, FE and RE, respectively) in (**C**), and *B. javanica* (leave, fruit, root) extracted by water (LW, FW and RW, respectively) in (**D**). The error bars are represented as the standard deviation (SD) of the mean from the triplicate experiment.

**Figure 2 pathogens-14-00646-f002:**
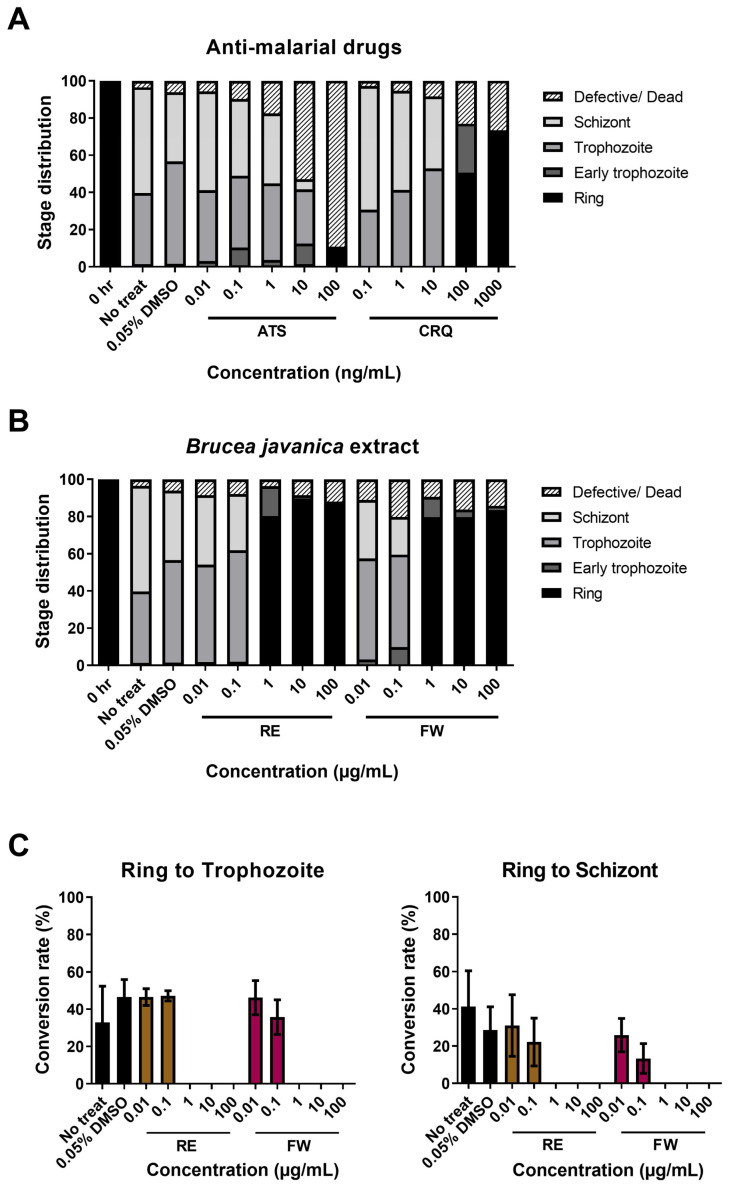
Distribution of parasite stages after *B. javanica* extracted treatments. The bar graph shows the proportion of ring, early trophozoite, trophozoite, schizont, and defective/death parasites counted under the microscope. Stage distribution of parasites treated by antimalarial drugs, including ATS and CRQ (**A**), and plant extracts, including RE and FW (**B**). The numbers of treated parasites by RE (brown bars) or FW (pink bars) were also calculated as a conversion rate from ring to early trophozoite and ring to schizont compared to the negative controls, including untreated (RPMI alone) and 0.05% DMSO which were represented by the black bars (**C**).

**Figure 3 pathogens-14-00646-f003:**
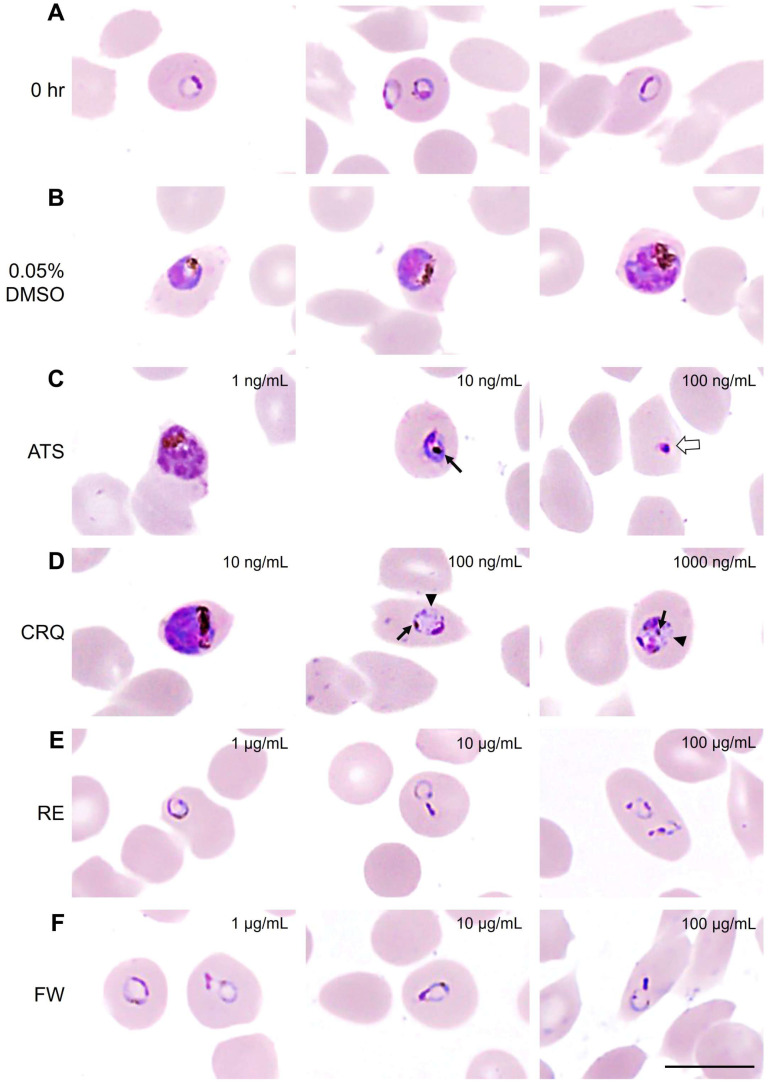
Photomicrographs of malaria parasite morphology after treatment with antimalarial drugs and plant extracts. The photomicrograph represents typical ring stage parasites at 0 h (**A**) and the early trophozoite, trophozoite, and schizont (**B**; left to right) from the negative control group treated with 0.05% DMSO. The parasite images from the group of positive controls, including artesunate treatment (**C**) and chloroquine treatment (**D**), were illustrated together with those of the parasites treated with root extract obtained by ethyl acetate (**E**) and the fruit extract obtained by water (**F**). The treatment dosages were indicated with the images. Nuclear clumping (white arrow); Hemozoin clumping (black arrow); vacuole accumulation (arrowhead). The scale bar indicates 10 µm.

**Figure 4 pathogens-14-00646-f004:**
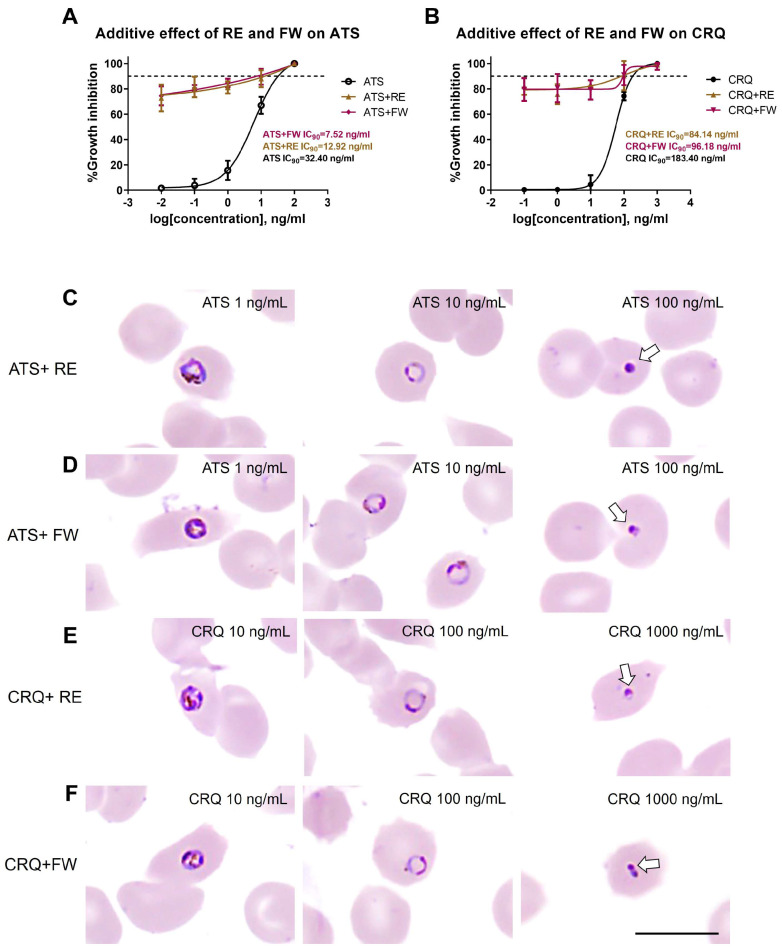
Additive effects of *B. javanica* extracts on antimalarial drugs. (**A**,**B**) represent the dose-response curves demonstrating the relationship between the logarithmic scale concentration of antimalarial drugs used and % growth inhibition of the RE and FW extracts on ATS in A and those on CRQ in B, compared with the antimalarial drugs alone—the error bars represent the standard deviation (SD) of the mean of a triplicate experiment. (**C**–**F**) represent the light micrographs illustrating the morphological changes of the parasite after additive treatments of RE and FW on ATS (ATS+RE and ATS+FW, respectively) and those on CRQ (CRQ+RE and CRQ+FW, respectively). The parasites were arrested at the ring and early trophozoite stages, as 0.01 ng/mL ATS and 0.01 ng/mL CRQ were treated with plant extracts. However, the figure revealed only 1, 10, and 100 ng/mL ATS and 10, 100, and 1000 ng/mL CRQ. The morphological change, characterized by nuclear clumping (indicated by white arrows), was observed in the early trophozoites at concentrations of 1 ng/mL of ATS+RE and 10 ng/mL of ATS+FW, respectively, in (**C**,**D**). The additive treatments on 10 ng/mL CRQ in (**E**) and (**F**) also displayed nuclear clumping (white arrows) in the trophozoites. The parasite was mainly in the ring stage after RE and FW treatments with 100 ng/mL CRQ. The dead parasites (indicated by the white arrow) were revealed after additive treatments of RE and FW at 100 ng/mL ATS (**C** and **D**, respectively) and those at 1000 ng/mL CRQ (E and F, respectively). (Scale bar = 10 µm).

## Data Availability

The data presented in this study are available upon request from the corresponding author, as they are interconnected with an ongoing project.

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
