# Peer review of "Growth Inhibition and Additive Effect to Antimalarial Drugs of Brucea javanica Extracts on Asexual Blood-Stage Plasmodium falciparum"

_pathogens, 2025, doi:10.3390/pathogens14070646_

Round 1
Reviewer 1 Report
Comments and Suggestions for Authors
This is a preliminary study on the antimalarial activity of B. javanica extracts on P. falciparum intraerythrocytic stages. Although the use of a biochemical assay to demonstrate antimalarial activity would have increased the solidity of the results, I think that the paper is worth being published.
However, I suggest both major and minor revisions.
Major revisions:
Hemolysis was tested to select non-toxic doses of extracts. However, cytotoxicity data on a human cell line would have been needed. If this is not possible, the authors should at least cite available evidence of no toxicity of these same extracts on human cells
The hemolysis method should be briefly described (line 135).
How the conversion rate from rings to trophozoites was calculates? Please add in the method section. Why the conversion rate for control drugs was not presented in the results section?
Additive (throughout the text) or synergistic (line 445)? This is confusing. Moreover, authors should specify which method has been used to define the interaction between the extract and the drug?
The IC50 should be discussed compared to the activity shown by other plant extracts. Micromolar IC50 are not low enough compared to active principles, but are quite normal for plan extracts. Authors should clarify this to the reader.
IC50 of chloroquine in Figure 1 and 4 is very different. May you explain why?
The results of statistical analysis are not adequately presented on the graphs.
Supplementary data file should be re-organized. Too many methodological details and comment on the data are present in the figure legends. Please add a method section to describe PCR and transfer the discussion on the different results from microscopic analysis and PCR in the main text at the end of paragraph 3.1.
Figure S2 present interesting data which authors should include in the main text.
Legend of figure 3 is not well written (line 299-300 It is not clear why that question is there; line 293: “represented; please make the legend uniform and schematic. Remove data discussion to move it in the text.
Minor revisions:
Some adjectives throughout the text are subjective, whereas data need to be precisely defined. Examples: line 28: strongly inhibited; line 216 “high level”; line 450 “satisfactory level”,
Check the spelling and wording throughout the text: Subscript the 50 in IC50 (examples: line 21, 310, 395); Names of microorganisms, mosquitos and plants in italics (examples: line 16, 360, 361-362, 419)
Please, don’t use “malaria” to mean “malaria parasites”. This has been done especially in the discussion section (examples: line 416, line 417, 423, 425, 426, 442)
Line 11: “endemic” where? I propose to change in “Malaria is a parasitic infectious disease that is endemic in many tropical countries” or something similar.
Line 12-13: “the incidence of treatment failure resulting in neurological conditions and death has remained” neurological conditions and death are not due to treatment failure only. Please clarify the sentence.
Line 14: what is the difference between agents and compounds? Please better explain.
Line 17: in this experiment (why singular?). I deduced that the experiment was repeated 3 times. Is this correct? Has the morphologic study been performed at least twice?
Line 20: optimal why?
Line 50: asexual intraerythrocytic. Please specify since also the intrahepatic cycle is asexual but it not symptomatic.
Line 55-60: why is this sentence in the past tense?
Line 61-69: please describe CQ resistance first and then ART resistance for chronological reasons.
Line 130: under microscope after Giemsa staining
Line 149-153: please reduce this part since there is a specific paragraph later on.
Line 200: please check the sentence
Line 204: is there a reason for this precise time? 26-28 hours? If yes, please clarify. Other ways a generic “24h” would be fine.
Line 220: “higher quantity”. What the authors do mean? Please better explain.
Line 235-236: I do not understand the meaning of this sentence.
Ine 237: could reveal (without a final “ed”)
Line 242-243: are authors sure of this interpretation? The retardation of development is due to less inhibitory activity?
Line 245: after treatment
Line 320-322: since authors used the IC50 dose, please better comment these results.
Line 4121: it depends on what drug you are referring to. Please specify.
Line 423-424: “larger hemozoin” is not clear/correct. Please change.
Line 443-444: “by existing synergistic activities” is not clear. Please, rephrase
Line 450-451: please check this sentence “The plant extract-treated parasites clearly demonstrated an irreversible effect on parasites arrested at the ring stage.” Parasites demonstrated an effect on parasites?
Please, rename figures in supplementary data section. Figure S1a and S1b are not linked, they should become S1 and S2. Then rename the following figures accordingly
Check references:
Line 54: reference number 5 and 6 seems too specific and not focused on the subject of the sentence.
Line 70: a review would be better choice for citation at this point.
Line 73: a reference for the following sentence is missing: One of the promising sources of new therapeutics is traditional medicinal plants.
Line 79: ref 23 is about the effect on cancer, too
Line 87-88: please cite the papers suggesting “a potential synergistic effect with chloroquine”,
Line 369: missing “]”
Line 380: [35-38]
Line 569 (ref number 37): please check the characters
Line 571-572 (ref number 38): please check
Line 461-462: please remove the first sentence in the Author Contributions section
Line 474 (acknowledgement): correct the sentence as follow: “We would like to thank all the staff in the Department of Preclinical Science”
Author Response
A point-by-point response to the reviewer’s comments
Reviewer 1
Major revisions:
Comment 1: Hemolysis was tested to select non-toxic doses of extracts. However, cytotoxicity data on a human cell line would have been needed. If this is not possible, the authors should at least cite available evidence of no toxicity of these same extracts on human cells
Response: Thank you very much for your suggestion. We inserted a context with a citation on the cytotoxic effect of B. javanica fruit extract on peripheral blood mononuclear cells, together with HepG2, a human hepatocellular carcinoma cell line, into the discussion section. This information explained the toxicity and safety dose that could be compared to the effect of plant extracts mentioned in our experiment. (lines 444-448)
Comment 2: The hemolysis method should be briefly described (line 135).
Response: A brief hemolysis method was included in Section 2.4 of the Materials and Methods. (lines 144-147)
Comment 3: How the conversion rate from rings to trophozoites was calculates? Please add in the method section. Why the conversion rate for control drugs was not presented in the results section?
Response: We added a sentence explaining the calculation of the conversion rate in Materials and Methods 2.6. From the parasite stage distribution represented in the figure 2A, the graph of ATS and CRQ treated groups did not show a clear pattern of ring stage arrest when compared to that observed in RE and FW treated groups (Figure 2B), therefore the conversion rate was only presented for the effect from plant extracts. (line 198-201)
Comment 4: Additive (throughout the text) or synergistic (line 445)? This is confusing. Moreover, authors should specify which method has been used to define the interaction between the extract and the drug?
Response: We changed the word “synergistic” to “additive” and defined this interaction in Result 3.3. From our study, we observed the effect of plant extracts on blocking parasite development at the ring stage without any morphological damage (e.g., vacuolization and hemozoin clumping) that is commonly found in ART or CRQ treatment. We implied that the plant extract obstructed the parasite growth in different partway of ART or CRQ. Therefore, the word additive is suitable for our investigation. (line 311-314)
Comment 5: The IC50 should be discussed compared to the activity shown by other plant extracts. Micromolar IC50 are not low enough compared to active principles, but are quite normal for plant extracts. Authors should clarify this to the reader.
Response: The microgram level of IC50 revealed by our plant extracts has been clarified in the discussion part as follows:
“Although the antimalarial effect of RE and FW in this experiment was revealed at 20-100 times lower concentrations than those of CRQ and ART, which showed their inhibitory effect at the nanogram level, the IC50 values of RE and FW, which were in the sub-microgram range, could be classified as excellent activity. Among 756 herb extracts reviewed for in vitro antiplasmodial activities, B. javanica extracted by methylene chloride was categorized in the group of highest activity together with the crude extracts from Ha-rungana madagascariensis and Quassia africana, and pure compounds established from Picrolemma spruce, Aspidosperma vargasi, Aspidosperma desmanthum, and Artemisia annua [47]”. (line 421-429)
Comment 6: IC50 of chloroquine in Figure 1 and 4 is very different. May you explain why?
Response: The parameter shown in Figure 1 was IC50, but the one presented in Figure 4 was IC90. In Figure 1, we focused on the inhibitory effect of plant extracts in comparison to ART or CRQ; therefore, we represented the effective doses as IC50 values. With these IC50 values, we can design a combination treatment (Figure 4) between plant extracts (by fixing the dosage at the IC50) and antimalarial drugs (by varying the dosage from 0.01 to 100 ng/mL for ATS and 0.1 to 1,000 ng/mL for CRQ). Due to the strong inhibitory effect of both groups of combination treatments (extract + ART and extract + CRQ) that could reach to 70% growth inhibition from the lowest dose of ART (0.01 ng/mL) and CRQ (0.1 ng/mL), the data could not be calculated into a reliable IC50 value. Therefore, the parameter to measure the efficiency of the combination treatment was presented by IC90 for demonstration of the inhibitory effect of the combination treatment.
To make it clear for this point, we added a sentence into the results part 3.4 related to figure 4 as “Since the inhibitory effect of the combination treatment dramatically increased to more than 70% from the lowest dose of antimalarial drugs, the inhibitory effect was represented by the IC90 value.” (line 333-335)
Comment 7: The results of statistical analysis are not adequately presented on the graphs.
Response: Since all the assays were performed in triplicate experiments. The graphs were shown with error bars representing the standard deviation (SD) of the data. There were only Figures 2A and B, in which the parasite stage distribution was presented in proportion; therefore, the SD was not demonstrated.
Comment 8: Supplementary data file should be re-organized. Too many methodological details and comment on the data are present in the figure legends. Please add a method section to describe PCR and transfer the discussion on the different results from microscopic analysis and PCR in the main text at the end of paragraph 3.1.
Response: We modified the figure legends in the supplementary data and reorganized the data explanation from Figure S2 to the end of the last paragraph in Section 3.1 (lines 236-242). (in supplementary data file and line 214, 234, and 287)
Comment 9: Figure S2 present interesting data which authors should include in the main text.
Response: From our purpose, the result from the molecular technique in Figure S2 was to support the main microscopic technique used throughout the entire experiment. Therefore, we would like to present this result in supplementary data as the original. However, the figure legend of Figure S2, which is related to the data explanation, was modified and reorganized into the main text 3.1. (line 231-236)
Comment 10: Legend of figure 3 is not well written (line 299-300 It is not clear why that question is there; line 293: “represented; please make the legend uniform and schematic. Remove data discussion to move it in the text.
Response: Thank you very much for your suggestion. We’ve found the mistake. The legend of Figure 3 has been rewritten. (line 320-327)
Minor revisions:
Comment 11: Some adjectives throughout the text are subjective, whereas data need to be precisely defined. Examples: line 28: strongly inhibited; line 216 “high level”; line 450 “satisfactory level”,
Response: We removed the adjectives as the reviewer suggested.
Comment 12: Check the spelling and wording throughout the text: Subscript the 50 in IC50 (examples: line 21, 310, 395); Names of microorganisms, mosquitos and plants in italics (examples: line 16, 360, 361-362, 419)
Response: Thank you for pointing this out. We have carefully reviewed the entire manuscript and revised all instances of “IC50 and IC90” to use the correct subscript formatting (ICâ‚…â‚€ and IC90). In addition, we have italicized all scientific names of Plasmodium falciparum, Anopheles mosquito, and Brucea javanica throughout the text as appropriate. The suggested examples have been corrected, and the formatting has been consistently applied throughout the manuscript.
Comment 13: Please, don’t use “malaria” to mean “malaria parasites”. This has been done especially in the discussion section (examples: line 416, line 417, 423, 425, 426, 442)
Response: We have revised the text, especially in the discussion section and throughout the manuscript, to ensure that the term “malaria” is not used to refer to the parasites. In the indicated lines and other lines, we have replaced “malaria” with “malaria parasites” or “parasites” as appropriate to improve clarity and accuracy. (line 453, 454, 461, 465, 479)
Comment 14: Line 11: “endemic” where? I propose to change in “Malaria is a parasitic infectious disease that is endemic in many tropical countries” or something similar.
Response: We have revised the sentence to clarify the geographic context as recommended. The sentence now reads: “Malaria is a parasitic infectious disease that is endemic in many tropical countries.” (line 11-12)
Comment 15: Line 12-13: “the incidence of treatment failure resulting in neurological conditions and death has remained” neurological conditions and death are not due to treatment failure only. Please clarify the sentence.
Response: We agree that neurological complications and death can arise not only from treatment failure but also from the severity of infection and drug resistance. We have revised the sentence for clarity and accuracy. (line 12-14)
Comment 16: Line 14: what is the difference between agents and compounds? Please better explain.
Response: To improve clarity, we have revised the sentence to distinguish between “therapeutic agents” (which refers to substances used as treatments) and “chemical compounds” (which refers more broadly to potential molecules under investigation, including those not yet approved as drugs). (line 14-16)
Comment 17: Line 17: in this experiment (why singular?). I deduced that the experiment was repeated 3 times. Is this correct? Has the morphologic study been performed at least twice?
Response: We appreciate the reviewer’s careful reading and valuable feedback. (line 18-21)
Regarding the use of "this experiment" (singular):
The singular form was used because the sentence refers to the overall experimental approach (treating parasites with plant extracts via single/co-incubation methods) rather than individual replicates. However, we acknowledge that this could be clearer. We will revise the phrasing to:
"In this study, malaria parasites were treated with plant extracts using single and co-incubation methods with artesunate/chloroquine..."
Repetition of experiments:
Yes, the experiment was independently repeated three times (biological replicates), with morphological analysis performed for each replicate. We have already clarified this in the Method Section 2.4 (Antimalarial activity test):
- Section 2.4 (Antimalarial activity test):
"Each treatment condition was performed in duplicate wells of three independent experiments…..." - Section 2.6 (Microscopic analysis):“In each treatment condition, five independent slides were counted for 10,000 erythrocytes……."
Comment 18: Line 20: optimal why?
Response: We appreciate the reviewer’s question. In this context, "optimal doses" refer to the most effective concentrations of plant extracts that demonstrate antimalarial activity (i.e., inhibit Plasmodium growth) while remaining non-toxic to red blood cells (RBCs). To improve clarity, this sentence was revised. (line 21-22)
Comment 19: Line 50: asexual intraerythrocytic. Please specify since also the intrahepatic cycle is asexual but it not symptomatic.
Response: We would like to thank the reviewer for this necessary clarification. To avoid uncertainty, we have revised the text to specify that the symptomatic phase of malaria is driven by asexual intraerythrocytic replication (not the asymptomatic liver stage). The updated sentence to " The repeated rounds of asexual intraerythrocytic replication led to clinical symptoms such as high fever with severe shaking chills and anemia, contributing to disease severity." (line 52-54)
Comment20: Line 55-60: why is this sentence in the past tense?
Response: All of the sentences have been revised. (line 59-62)
Comment 21: Line 61-69: please describe CQ resistance first and then ART resistance for chronological reasons.
Response: (lines 65-74) We have reorganized the paragraph to present drug resistance chronologically, starting with chloroquine (CQ) resistance and then discussing artemisinin (ART) resistance. Below is the revised version:
Revised text:
" The emergence and spread of drug-resistant P. falciparum strains have become a significant obstacle to malaria control and elimination [10,11]. Chloroquine (CRQ) was once the primary malaria drug, but P. falciparum developed widespread resistance. However, it remains effective against P. vivax and P. malariae in certain regions [12,13]. Artemisinin-based combination therapy (ACT), which pairs fast-acting artemisinin derivatives (e.g., artesunate, dihydroartemisinin) with longer-acting partner drugs (e.g., piperaquine, lumefantrine, mefloquine), was subsequently developed as the first-line treatment [14,15]. However, partial resistance to artemisinin and decreasing efficacy of partner drugs, especially in the Greater Mekong Subregion, including Thailand, Cambodia, and Myanmar [16-19].
Comment 22: Line 130: under microscope after Giemsa staining
Response: We have revised the sentence to clarify that parasite stage confirmation was performed under the microscope after Giemsa staining. (line 139-141)
Comment 23: Line 149-153: please reduce this part since there is a specific paragraph later on.
Response: Thank you for the suggestion. We have condensed the content in this section to avoid redundancy and now provide a brief summary of the methodology. (line 156-161)
Comment 24: Line 200: please check the sentence
Response: We have revised the sentence to “From the hemolysis assay, the leaf and root extracts prepared with ethyl acetate (LE and RE) at 1,000 μg/mL were found to be toxic to normal red blood cells.” (line 210-212)
Comment 25: Line 204: is there a reason for this precise time? 26-28 hours? If yes, please clarify. Other ways a generic “24h” would be fine.
Response: The 26-28 hours time point was intentionally selected to ensure that most parasites had progressed beyond the ring stage and reached the trophozoite or schizont stages, which are more morphologically distinct for assessing growth inhibition. Since the culture was initiated with synchronized ring-stage parasites at 6–12 hr post-invasion, a 24-hour incubation may not have allowed sufficient development to observe later-stage morphology in all parasites. We have retained the 26–28 hr time point and added clarification in the text. (line 216-219)
Comment 26: Line 220: “higher quantity”. What the authors do mean? Please better explain.
Response: We have clarified that “higher quantity” refers to relatively elevated levels of 18S rRNA expression detected by real-time RT-PCR, likely reflecting the higher sensitivity of this molecular approach compared to microscopic analysis. (line 231-236)
Comment 27: Line 235-236: I do not understand the meaning of this sentence.
Response: We changed the sentence as mentioned in lines 256-258.
Comment 28: Line 237: could reveal (without a final “ed”)
Response: As per the above comment, the sentence has been modified, and the typo has been corrected. (line 256-258)
Comment 29: Line 242-243: are authors sure of this interpretation? The retardation of development is due to less inhibitory activity?
Response: We have clarified the sentence to better reflect our observation. While CRQ exhibited limited inhibitory activity, it still caused a delay in parasite development, with parasites remaining at the ring and early trophozoite stages and showing minimal signs of death. (line 263-266)
Comment 30: Line 245: after treatment
Response: We have revised the sentence for grammatical accuracy. (line 266-268)
Comment 31: Line 320-322: since authors used the IC50 dose, please better comment these results.
Response: We agree that a more straightforward interpretation is needed regarding the use of ICâ‚…â‚€ concentrations of the plant extracts. We have revised the paragraph to emphasize that RE and FW, when administered at their ICâ‚…â‚€ doses, significantly enhance the efficacy of low-dose ATS and CRQ, resulting in 70–80% growth inhibition. This outcome strongly supports the additive effect of B. javanica extracts in enhancing antimalarial activity even at subtherapeutic drug concentrations. (line 341-347)
Comment 32: Line 421: it depends on what drug you are referring to. Please specify.
Response: We have revised the sentence to specify the drugs being referred to. Since our study focused on artesunate and chloroquine, we now clarify that the ring stage is susceptible to these compounds. (line 456-460)
Comment 33: Line 423-424: “larger hemozoin” is not clear/correct. Please change.
Response: We have changed from “large hemozoin” to “a considerable accumulation of hemozoin”. (line 461)
Comment 34: Line 443-444: “by existing synergistic activities” is not clear. Please, rephrase
Response: We agree that the phrase “by existing synergistic activities” was unclear. We have revised the sentence to clarify that the additive effects were observed in our study. (line 482-485)
Comment 35: Line 450-451: please check this sentence “The plant extract-treated parasites clearly demonstrated an irreversible effect on parasites arrested at the ring stage.” Parasites demonstrated an effect on parasites?
Response: The revised sentence is: “The plant extracts demonstrated irreversible developmental arrest at the ring stage, indicating a potent and lasting effects on the parasites.” (line 491-493)
Comment 36: Please, rename figures in supplementary data section. Figure S1a and S1b are not linked, they should become S1 and S2. Then rename the following figures accordingly
Response: We have changed the names of the figures in the supplementary data to Figure S1, S2, and S3, respectively. (in supplementary data file and lines 214, 234, and 286)
Check references:
Line 54: reference number 5 and 6 seems too specific and not focused on the subject of the sentence.
Response: We added more scientific information to relevant with references. (line 54-58)
Line 70: a review would be better choice for citation at this point.
Response: We modified this part by adding more information on artemisinin-resistant parasites, including the citation on review articles. (line 72-79)
Line 73: a reference for the following sentence is missing: One of the promising sources of new therapeutics is traditional medicinal plants.
Response: The reference numbers 20 and 21 are indicated in this sentence. (line 78)
Line 79: ref 23 is about the effect on cancer, too
Response: The sentence has been modified, and ref . 25 has been added. (line 88-89)
Line 87-88: please cite the papers suggesting “a potential synergistic effect with chloroquine”,
Response: The ref number 37 has been cited.(line 97)
Line 369: missing “]”
Response: The “]” has been added. (line 392)
Line 380: [35-38]
Response: It has been changed to [37-40]. The number of references has been updated due to modifications in the previous section. (line 403)
Line 569 (ref number 37): please check the characters
Response: The text characters have been changed to sentence case (It is now ref number 39). (line 609)
Line 571-572 (ref number 38): please check
Response: The format has been corrected (It is now ref number 40). (line 610)
Line 461-462: please remove the first sentence in the Author Contributions section
Response: The guiding sentences have been removed. (line 499-505)
Line 474 (acknowledgement): correct the sentence as follow: “We would like to thank all the staff in the Department of Preclinical Science”
Response: We’ve corrected the sentence as the reviewer suggested. (line 509)

Reviewer 2 Report
Comments and Suggestions for Authors
Growth inhibition and additive effect to antimalarial drugs of Brucea javanica extracts on asexual blood-stage Plasmodium falciparum
The authors of this article focus on the antiplasmodial activity of a plant used in folk medicine, mainly in Asia, for various pathologies. The anti-parasitic properties of Brucea javanica extracts against Plasmodium falciparum and Plasmodium berghei have already been demonstrated, but in a relatively succinct manner. The article provides clear details on the impact of the extracts tested on the development of the intra-erythrocytic cycle of P. falciparum, alone or in combination with artesunate or chloroquine.
The introduction sets the subject of the study in the global context of malaria, which is predominantly found in sub-Saharan Africa. It should be pointed out that artemisinin resistance is not confined to South-East Asia, but has become a major concern since its recent emergence and spread in Africa.
The material and methods section is sufficiently detailed, except for the microscopic analysis, for which it would be useful to specify whether the counts were made by a single person and the slides were counted several times.
Regarding parasite synchronization, a single sorbitol treatment generally leads to 0-20 hr rings, while obtaining 0-12 hr rings requires 2 sorbitol treatments. Is this the case?
No particular comments on the results section. A scale should be added to the photos in figures 3 and 4 to estimate the size of the infected red blood cells.
A few nuances regarding the discussion. Despite convincing in vitro results on the antiplasmodial activity of Brucea javanica extracts, particularly in combination with artesunate, we should insist a little more on the need to calibrate extracts in terms of active ingredient content and antiparasitic activity (this is a little too briefly mentioned).
Note that Brucea javanica extracts do not kill malaria but parasites! Same remark in the following lines: it's not malaria that's directly concerned, but P. falciparum!
Several typographical errors such as IC50 or IC90 instead of IC50 and IC90
In the abstract and line 56, P. falciparum is not italicized.
In conclusion, the article provides new information on the effect of Brucea javanica extracts on the development of P. falciparum in red blood cells.
Author Response
A point-by-point response to the reviewer’s comments
Reviewer 2
Comment 1: The authors of this article focus on the antiplasmodial activity of a plant used in folk medicine, mainly in Asia, for various pathologies. The anti-parasitic properties of Brucea javanica extracts against Plasmodium falciparum and Plasmodium berghei have already been demonstrated, but in a relatively succinct manner. The article provides clear details on the impact of the extracts tested on the development of the intra-erythrocytic cycle of P. falciparum, alone or in combination with artesunate or chloroquine.
The introduction sets the subject of the study in the global context of malaria, which is predominantly found in sub-Saharan Africa. It should be pointed out that artemisinin resistance is not confined to South-East Asia, but has become a major concern since its recent emergence and spread in Africa.
Response: (line 74-79) We inserted the information on artemisinin resistance in Africa into the paragraph of introduction with references as follows:
“Moreover, artemisinin resistance in Africa was recently identified with the increasing prevalence of the PfKelch13 mutation. These artemisinin-resistant PfKelch mutants were found in low transmission in several countries, including Rwanda, Uganda, Ethiopia, Sudan, Kenya, and eastern DRC, showing the spread of artemisinin-resistant parasites over the large distances [20,21].”
Comment 2: The material and methods section is sufficiently detailed, except for the microscopic analysis, for which it would be useful to specify whether the counts were made by a single person and the slides were counted several times.
Response: We added the information into materials and methods 2.6. (line 197-198)
Comment 3: Regarding parasite synchronization, a single sorbitol treatment generally leads to 0-20 hr rings, while obtaining 0-12 hr rings requires 2 sorbitol treatments. Is this the case?
Response: Thank you for your question. Our synchronization protocol was done by two consecutive sorbitol treatments. The first was to eliminate late trophozoite and schizont stages, while the second helped eliminate residual early trophozoites, allowing for tighter synchronization. We have revised the materials and methods 2.3 accordingly for clarification. (line 138-139)
Comment 4: No particular comments on the results section. A scale should be added to the photos in figures 3 and 4 to estimate the size of the infected red blood cells.
Response: The scale bar for all images is the same in each Figure, and it is indicated in the last image at the lower right of both Figures 3 and 4.
Comment 5: A few nuances regarding the discussion. Despite convincing in vitro results on the antiplasmodial activity of Brucea javanica extracts, particularly in combination with artesunate, we should insist a little more on the need to calibrate extracts in terms of active ingredient content and antiparasitic activity (this is a little too briefly mentioned).
Response: We modified the last sentence of the discussion part to emphasize the impact of further on the active ingredient of this plant. (line 482-485)
Comment 6: Note that Brucea javanica extracts do not kill malaria but parasites! Same remark in the following lines: it's not malaria that's directly concerned, but P. falciparum!
Response: Thank you for your suggestion. All are corrected.
Comment 7: Several typographical errors such as IC50 or IC90 instead of IC50 and IC90
Response: We have carefully reviewed the entire manuscript and revised all instances of “IC50 and IC90” to use the correct subscript formatting (ICâ‚…â‚€ and IC90).
Comment 8: In the abstract and line 56, P. falciparum is not italicized.
Response: Thank you for pointing this out. We have italicized all scientific names of Plasmodium falciparum, Anopheles mosquito, and Brucea javanica throughout the text as appropriate. The suggested examples have been corrected, and the formatting has been consistently applied throughout the manuscript.
Comment 9: In conclusion, the article provides new information on the effect of Brucea javanica extracts on the development of P. falciparum in red blood cells.
Response: Thank you for your valuable comment.

Round 2
Reviewer 1 Report
Comments and Suggestions for Authors
The authors answered to my questions and I think that now the manuscript is improved and suitable for publication.
The only exception is this comment, probably it was not clearly written from my part:
Supplementary data file should be re-organized. Too many methodological details and comment on the data are present in the figure legends. Please add a method section to describe PCR and transfer the discussion on the different results from microscopic analysis and PCR in the main text at the end of paragraph 3.1.
Authors have removed comments on data but not methods. Why authors did not introduce a methods section (with PCR details) in the supplementary file, before supplementary figures? Methodological details are not supposed to be find in the figure legends, where I expect to find few details only, relevant to understand the data reported in the figure.
Minor revisions:
Line 251-252: please check, shouldn’t that be a single sentence?
My only concern is about the numbering of referencing. A final check should be performed. For instance, Reference 47 and 48 in the text seems to be inverted in the reference list.
Line 428: the correct reference is “47. Lu, YH; Yang, YY; Pan, IH. Anti-proliferative Effect of Brucea javanica Fruit Extract Against Human Hepatocarcinoma Cell 608 Lines and Its Mechanism. Planta Med 2011, 77, PM56, doi: 10.1055/s-0031-1282814.”, ins’t it?
Line 410: the correct reference is “48. Lemma, M.T.; Ahmed, A.M.; Elhady, M.T.; Ngo, H.T.; Vu, T.L.; Sang, T.K.; Campos-Alberto, E.; Sayed, A.; Mizukami, S.; Na- 610 Bangchang, K.; Huy, N.T.; Hirayama, K.; Karbwang, J. Medicinal plants for in vitro antiplasmodial activities: A systemic review 611 of literature. Parasitol Int 2017, Dec;66(6):713-720. doi: 10.1016/j.parint.2017.09.002”, ins’t it?
Author Response
A point-by-point response to the reviewer’s comments
Reviewer 1
Comment 1: Supplementary data file should be re-organized. Too many methodological details and comment on the data are present in the figure legends. Please add a method section to describe PCR and transfer the discussion on the different results from microscopic analysis and PCR in the main text at the end of paragraph 3.1.
Authors have removed comments on data but not methods. Why authors did not introduce a methods section (with PCR details) in the supplementary file, before supplementary figures? Methodological details are not supposed to be find in the figure legends, where I expect to find few details only, relevant to understand the data reported in the figure.
Respond 1: Thank you very much for your suggestion. The supplementary data is now reorganized as per your recommendations.
Minor revisions:
Comment 2: Line 251-252: please check, shouldn’t that be a single sentence?
Respond 2: Thank you very much for your suggestion. One sentence has been removed. (lines 254-255)
Comment 3: My only concern is about the numbering of referencing. A final check should be performed. For instance, Reference 47 and 48 in the text seems to be inverted in the reference list.
Respond 3: Thank you very much for your valuable comment. References no. 47 and 48 have been re-checked and are now in the correct places.
